# The Social Stress Indicator (SSI)

**Herkulaas Combrink**[1,2]*

**1** Interdisciplinary Centre for Digital Futures/University of the Free State, Bloemfontein, Free State, South Africa, **2** Office of the Dean Economics and Management Sciences/University of the Free State, Bloemfontein, Free State, South Africa

* combrinkhm@ufs.ac.za

**Data availability statement:** Data relevant to this paper are available at https://doi.org/10.38140/ufs.28015310.v2.

## Abstract

Social media shapes perceptions, influences emotions, and fuels anxieties, yet its true impact on social stress remains difficult to measure. This study introduces the Social Stress Indicator (SSI) as a computational tool for quantifying social stress in real time by integrating sentiment analysis (SA), subjectivity (SUB), and information seeking behavior (ISB). The data used was synthetic, and no-real world data was used to test the context. To validate SSI, it was compared to an expert-labeled Social Stress Score (SSS), which assesses sentiment negativity, anxiety expression, engagement, misinformation, and help-seeking behavior. Results show that SSI moderately aligns with SSS but underestimates stress levels, especially during high-intensity events. Bland-Altman analysis confirmed a negative bias, suggesting SSI struggles to capture extreme stress. However, causality tests indicate SSI has predictive power, making it a potential early warning system for stress-related trends. Despite its promise, SSI faces limitations in detecting misinformation surges, crisis-driven anxiety, and nuanced social interactions. This study demonstrates the feasibility of computationally tracking social stress at scale but highlights the need for refinements. Future improvements should enhance sensitivity to extreme stressors, incorporate adaptive thresholding, and integrate contextual signals like network effects and linguistic nuances. This research advances infodemic intelligence, with implications for mental health monitoring, policy-making, and digital governance.

## Introduction

Information spreads fast, but so does stress, misinformation, and fear. If we can track viral trends, why not track their emotional impact? Social media platforms have become an integrated and widely used form of social interaction and communication [1]. According to a world social media statistics research summary 2024, more than half of the world uses or engages on social media [2,27]. Social media may be used privately between specific people, which is not open to the public, or publicly which may be viewed by anyone with an internet connection. When people engage on a public social media forum, it has the potential to disseminate information far and wide which may be problematic if the information is laden with misinformation, hate speech, or any other contributor towards social stress [3]. Social stress is a concept used in social psychology and other related disciplines that tries to measure

**Funding:** The author(s) received no specific funding for this work.

**Competing interests:** The authors have declared that no competing interests exist.

public stress, public harm, or the type of pressures placed on an individual that may lead to poor mental health outcomes from societal pressures [4]. While effective for capturing certain features of digital discourse, these approaches often fail to provide a comprehensive, integrated understanding of how social stress unfolds across time and topics in online environments. Existing models are limited in their ability to differentiate between transient emotional expressions and sustained social stress patterns that may signal broader societal disruption or mental health risk. Furthermore, many frameworks do not incorporate the temporal relevance of topics or the public's information-seeking behaviour—key factors in identifying the contextual significance and urgency of stress-inducing events [17,23]. This study addresses these critical gaps by developing and validating a Social Stress Indicator (SSI) that integrates sentiment analysis (SA), subjectivity (SUB), and information-seeking behaviour (ISB) to detect and track real-time manifestations of social stress on social media.

## Social media

Social media has become an area of interest around the world because of the marketing potential, business opportunities, and information dissemination potential [5]. Social media is also a form of socialization between members of the public in digital spaces using platforms to engage around a particular topic or within a particular discussion area around published media [6]. There are two types of engagements on social media: private engagements and public engagements. Private engagements are held between selected individuals in a closed system, and only the chosen participants have access to the dialogue. Public engagement on the other hand is conducted in virtual spaces where everyone with access to a network, such as the internet, can view the dialogue and potentially contribute toward the conversation [7]. Social media is seen as a two-way communication space where people may both send and receive information. However, social media alone is not enough to measure public discourse, but rather an addition to the discourse analysis of public dialogue. Social media data can have some biases due to an increase in social stigmatisation, either through social media users being more reluctant to share their true opinions or by being more radical to gain social approval [8]. Despite the biases, more users are engaging, and the rate of engagement is more rapid than before the 2020 global pandemic [9]. This means that the adoption and use of social media is widespread, global, and increasing. Social media, by its very nature, creates an ecosystem where users engage in continuous social comparison, ideological conflicts, and exposure to emotionally charged content [10].

## Social media and social stress

Social stress, a psychological and emotional burden experienced in response to social interactions and societal pressures, has become increasingly prominent in digital spaces [11]. Unlike traditional forms of social interaction, where stressors may be more immediate and localized, social stress on digital platforms is persistent, amplified, and far-reaching [12]. This stress emerges not only from direct interactions—such as cyberbullying, harassment, or negative feedback—but also from the broader influence of social norms, misinformation, performative behavior, and the fear of missing out (FOMO) [13]. At present, some social media platforms have policies regarding the type of content that it will allow—and if the content falls outside of the prescribed policy, the comment may be removed by the platform owners or content moderators, depending on the policy [14]. Additionally, some social media platforms may not have this type of content moderation, exacerbating the potential for harmful content which will in turn potentially increase social stress. Globally, there is a gap in research related to the

identification, monitoring and evaluation, and impact assessment of social media and social stress emanating from social media platforms [15].

A key driver of social stress in online environments is the pressure to conform [16]. Public engagement on social media often leads to self-censorship, performance-based interactions, and an excessive need for validation through likes, shares, and comments. This creates an inauthentic self-presentation, where users curate idealized versions of their lives, contributing to collective stress and unrealistic societal expectations. The need to engage, debate, or defend opinions in real-time, often in polarizing or hostile environments, further escalates stress levels among users. Moreover, misinformation, propaganda, and algorithm-driven content amplification contribute to collective anxiety, particularly in times of crisis or uncertainty [17]. While content moderation policies on social media platforms attempt to regulate harmful interactions, they often fail to capture the nuanced and evolving nature of social stressors. Sarcasm, passive-aggressive remarks, exclusionary behavior, and ideological attacks may fall outside explicit moderation guidelines yet still contribute significantly to social stress. Moreover, the anonymity and virality inherent in social media exacerbate these stressors, as users may feel emboldened to engage in hostile interactions without immediate consequences. The unfiltered, rapid-paced nature of public discourse can create a digital environment where stress compounds quickly, affecting both individuals and entire online communities [18].

Sentiment analysis (SA) and topic modeling (TM) have been used to measure online anxiety and stress [19]. On the one hand, SA deciphers the emotional tone of user-generated content, distinguishing between positive, negative, and neutral sentiments. This analysis provides insights into the collective emotional state of users, which is essential for assessing social stress levels. For instance, a study analyzing Reddit discussions used SA to gauge stress and anxiety expressions, highlighting its effectiveness in understanding user emotions in social media contexts [20]. On the other hand, TM identifies prevalent themes within large datasets. By uncovering dominant topics, researchers can pinpoint specific stressors or concerns affecting the online community. However, SA and TM alone do not provide evidence into the life cycle of a specific social stress concept. Given the complexity of social stress in digital spaces, traditional research methods have struggled to quantify its effects. The challenge lies in the scale of differentiating between temporary emotional reactions and sustained social stress patterns that contribute to poor mental health outcomes. While SA provides a general emotional snapshot, and TM helps identify dominant discourse themes, these methods alone are insufficient to track the life cycle of social stress [21]. To bridge this gap, social stress must be measured dynamically factoring in the frequency, intensity, and evolution of stress-related discussions over time. Given that social media is complex, and a form of public discourse, it remains challenging to study dialogue once the volume of a conversation reaches millions of interactions [22].

Another vital component of measuring social stress for social media is the incorporation of the information seeking behavior of people [23]. If a conversation is topical at that particular point in time, then the context of the conversation relevance can be added. If we can measure the economy in real time, why can't we measure the rising waves of social stress that affect millions daily? Therefore, the purpose of this study was to design and evaluate a social stress indictor for capturing the real-time manifestation of social stress on social media platforms.

## Materials and methods

### Research design

This study used a multistep methodological approach to develop and validate a Social Stress Indicator (SSI) by comparing it with an expert-labeled social stress score (SSS). The SSI

consisted of SA, SUB, and ISB. The selection of these wer theoretically and empirically grounded based on their demonstrated ability to capture distinct yet complementary aspects of social stress as manifested on social media platforms. SA is widely used in affective computing to infer users' emotional states from textual data, with negative sentiment consistently associated with heightened stress, anxiety, and distress in online communities [19,20]. SUB was included to measure the degree to which posts express personal opinions, beliefs, or perceptions—an essential proxy for emotional reactivity and interpretive bias, both of which are known stress amplifiers in digital discourse. Finally, ISB was incorporated to account for the temporal salience and societal relevance of stress-inducing topics. Drawing from information science and infodemiology, spikes in search interest or digital engagement with specific themes are often indicative of collective concern or uncertainty, particularly during crises or unfolding events [17,23]. Together, these three components were selected for their ability to provide a multi-dimensional, real-time representation of evolving stress narratives online. They serve not only as quantitative markers of affective and cognitive states but also as indicators of broader discursive trends, enabling the SSI to function as both a diagnostic and predictive tool for identifying stress-laden content across social media ecosystems. A synthetic data set containing 1000 low-stress, 1000 medium-stress and 1000 high-stress posts was used to conduct the experiments. To establish an independent ground truth measure, an expert labeled SSS was created using a rubric that evaluates each post in five dimensions: negative sentiment, anxiety expression, level of engagement, misinformation content and help seek behavior. Experts assign a score of 0-5 for each dimension, and the final SSS per post is calculated as the equally weighted average of these scores. The SSS is then aggregated over time windows to facilitate a direct comparison with the SSI. The threshold for low social stress was set to below 20% (0.2), moderate to 21%–40% and high above 40%.

## Ethics statement

All permission to use synthetic data and ethics for this study was obtained under the ethical clearance numbers UFS-HSD2020/1846/2601 and UFS-HSD2023/2095. Because the data was synthetic, informed consent was not needed, as the data was generated using a large language model. The prompts to generate the data encouraged that no demogrtaphic slurs or any demographic isolation was used. Additionally, each data point was evaluated by the researcher to ensure that no profanities of any kind featured within the data. Furthermore, data collection had a synthetic time stamp attached to it, to represent what real-world data could look like, but no real-world data were used, only synthetic data. The microblogs themselves did not contain profanity or hate speech. Data were generated in-house and no human participants were involved in the generation of data.

## Data

Synthetic microblogs were generated using ChatGPT based on prompting. The prompts focused on emulating 'social media like' microblogs, which means that the microblogs included hashtags as well as emojis. No human subjects were used in this study, and 3000 microblogs were generated using various prompts to ensure the uniqueness of the microblogs. All the microblogs contained a variety of different information. There were a different number of microblogs generated for each time stamp over a 7-day period for 24 hours per day. All the data is in a public repository [24]. In this study, no human data was used, only the synthetic data generated using a large language model.

## Social stress score

The SSS was designed to quantify social stress at the microblog level by evaluating key psychological and behavioral indicators and was adapted from the standard stress scale [25]. It was constructed using a rubric that scores each post across five dimensions: sentiment negativity, anxiety expression, engagement level, misinformation content, and help-seeking behavior, with each dimension rated on a scale from 0 to 5. The final SSS for a post is calculated as the equally weighted average of these scores, providing a standardized measure of social stress. Additionally, five expert raters trained in social sciences applied the rubric to synthetic microblogs generated to reflect diverse stress-inducing contexts. Each of the five experts independently scored each microblog and the average score was aggregated and used in the SSS rubric. The adaptation also included iterative refinements based on inter-rater reliability assessments, ensuring consistency and validity in measuring social stress across varying content types and linguistic expressions. The average SSS scores were used for the statistical comparisons between the SSS and SSI.

## Social stress indicator

The polarity of a person's emotions or opinions on a certain topic or event was measured using SA, as either positive (to a maximum value of 1 or 100%) or negative (to a minimum value of −1 or −100%) number. SA measure was thus defined as (Eq 1):

$$SA = -\sum_{I=0}^{N}\left(\frac{(x_1 + x_2 + x_3 + ... + x_n)}{N}\right),\qquad(1)$$

where SA is the sentiment, N is the sample size of the number of microblogs measured and $x_1, x_2, ..., x_n$ is the sentiment of each microblogs per time stamp. The average of the sentiment is then used as the SA for that time interval of the dataset. For this indicator there will thus be an average sentiment score ranging between −1 and 1 for each time stamp in the series. A social media lexicon was used for the SA. Subjectivity analysis was conducted using a lexical-based approach. In this study, each microblog was processed to extract subjective lexical units from a predefined subjectivity lexicon. For each microblog, the set $L$ denoted the top subjective words identified from the text. The subjectivity score, denoted by $SUB(t)$ for a given microblog $t$, was computed as follows (Eq 2):

$$\text{SUB}(t) = \frac{1}{|L|}\sum_{l_i \in L}\log\frac{S(l_i) + \epsilon}{1 - S(l_i) + \epsilon},\qquad(2)$$

where $S(l_i)$ represented the subjectivity rating of the word $l_i$ as provided by the lexicon (used in Text Blob and VADER), $|L|$ was the total number of subjective words identified in the microblog, and $\epsilon$ was a small smoothing constant added to avoid taking the logarithm of zero. The logarithmic ratio measured the relative strength of subjective language by comparing the probability of subjectivity to its complement (objectivity). This formulation was conceptually adapted from the coherence measure used in topic modeling where instead of assessing word co-occurrence, the focus was on the degree of subjectivity inherent in the text [26]. Subjectivity scores were calculated for each microblog individually, thereby quantifying the extent to which each text conveyed personal opinions and emotional content. These scores, in conjunction with sentiment polarity measures and other components, were averaged for each time interval for the overall indicator. ISB in the context of digital platforms like social media was defined as the measurement of the number of people that searched for a related

topic, over a certain period of time on that platform. The ISB measure was calculated as (Eq 3):

$$ISB = \frac{\sum_{i=0}^{N} \beta}{\sum_{i=0}^{N} \overline{X}}, \tag{3}$$

where $\beta$ is the frequency of searched topic at a specified time, and $\overline{X}$ is the highest frequency of the same topic at a specific point in time, expressed as a percentage. The measurement of ISB is always in the context of the series, and for this, Google trends were used. Once the SA, SUB, and ISB has was determined, the SSI indicator was calculated for each timestamp in the series. SSI was defined as (Eq 4):

$$SSI = \frac{-SA + (1 - SUB) + ISB}{3}, \tag{4}$$

where SA is seen as sentiment analysis (Eq 1) with a negative designation due to the emphasis on sentiment negativity, SUB is seen as subjectivity rating subtracted from 1 (Eq 2), and ISB is seen as information seeking behavior (Eq 3). In the case of the initial SSI indictor, the weightings are assumed as equal.

## Comparison between SSS and SSI

To compare the SSI and the expert-labeled SSS, Bland-Altman analysis and Granger causality testing were conducted. The Bland-Altman analysis assessed the level of agreement between the two measures by computing the mean bias, standard deviation of differences, and limits of agreement (LoA). This method was used to determine whether SSI and SSS were interchangeable in measuring social stress. Additionally, Granger causality tests were performed to examine whether past values of one variable could predict future values of the other, helping to identify whether SSI led or followed expert evaluations (SSS). Confidence intervals were calculated for both tests, with the Bland-Altman analysis focusing on measurement agreement and the Granger causality test identifying predictive relationships over time. In both cases, the ranges with the 95% confidence intervals were compared and not the absolute values.

## Results and discussion

For the SSI and SSS comparison with low stress microblogs, both indicators follow a similar trend, with SSI appearing to slightly underestimate SSS, particularly during peak stress periods. The confidence intervals overlap moderately, suggesting a level of agreement throughout the series. The Bland-Altman plot indicates a small negative bias (-0.025), where SSI slightly underestimates SSS on average. The LoA are within an acceptable range, with most points falling within these limits, which suggests reasonable agreement between the two measures for the entire series. The causality test results indicated that SSI has some predictive power for SSS, as a few points exceed the causality threshold (Fig 1).

For the medium stress dataset, the time-series comparison between SSI and SSS showed a more pronounced gap between the two measures, with SSI consistently underestimating SSS across almost the entire period (Fig 2). The confidence intervals barely overlap, and SSS remains higher than SSI throughout. The SSI was on average in the moderate threshold, but SSS was much higher. The Bland-Altman plot confirmed a negative bias of -0.193, indicating that the SSI is systematically underestimating stress compared to SSS. The limits of agreement are slightly wider than in the Fig 1, suggesting increased variability in how SSI aligns with SSS. Furthermore, the causality analysis in this version shows fewer points exceeding the causality

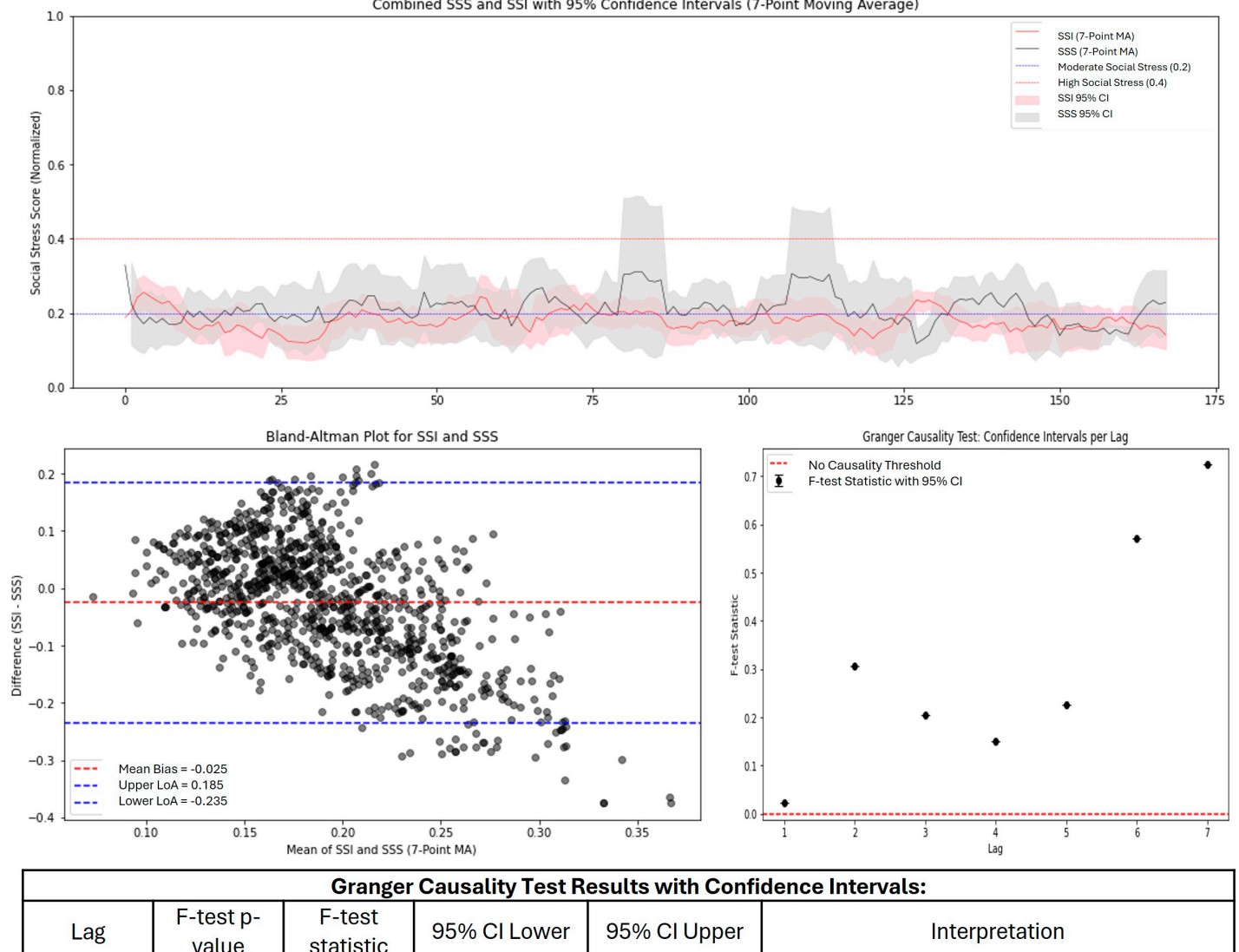

**Fig 1. SSI and SSS time series, Bland-Altman, and Granger causality for low stress microblogs.**

threshold, which means SSI is less effective at predicting SSS compared to the first evaluation. This version of SSI appears less reliable to the extent of the social stress, but could detect an increase in social stress as compared to the low stress dataset.

For the dataset containing high social stress, the confidence intervals no longer overlap significantly, indicating a mismatch in how the two measures assess social stress. The magnitude

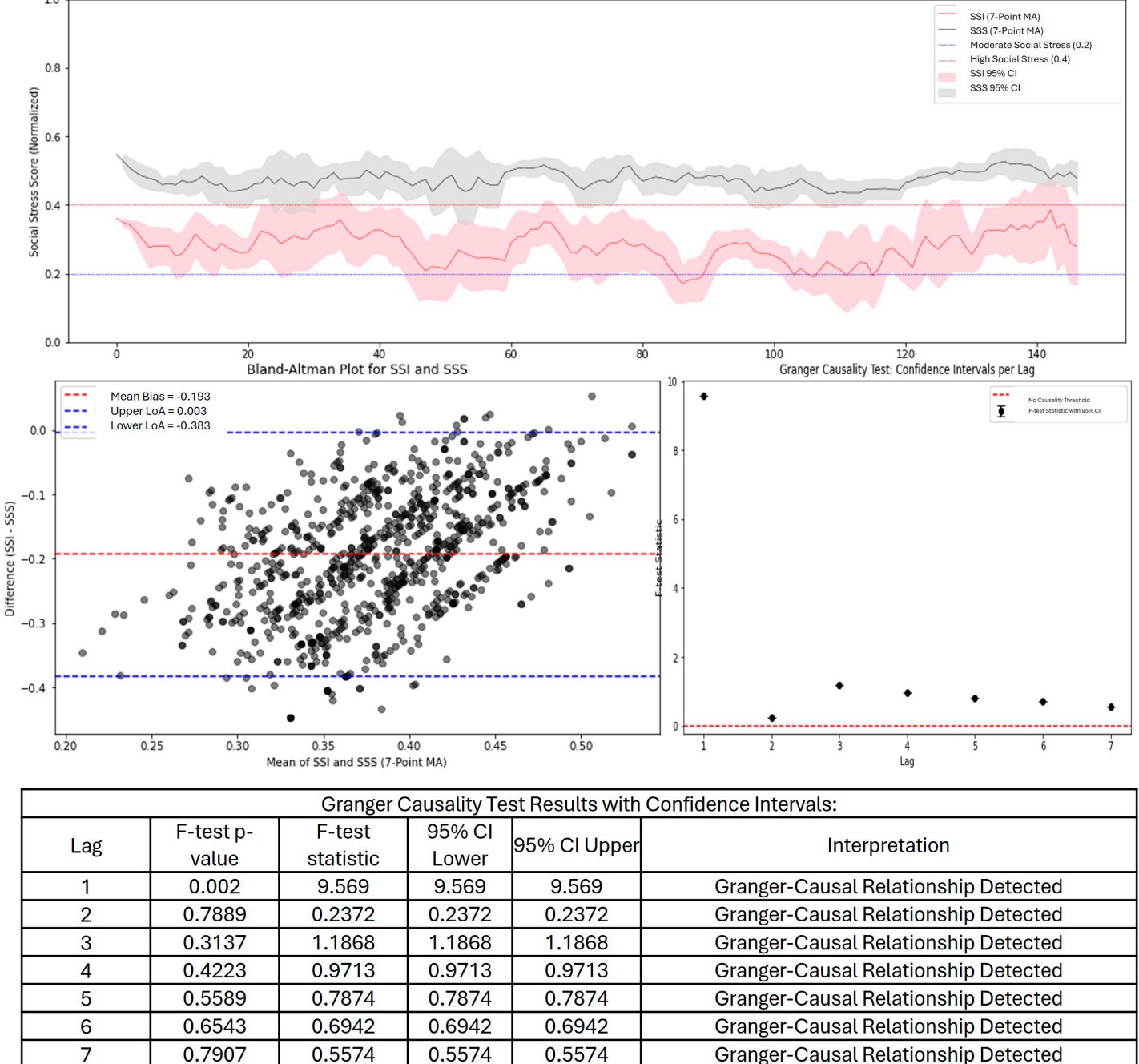

**Fig 2. SSI and SSS time series, Bland-Altman, and Granger causality for medium stress microblogs.**

of the underestimation has increased significantly compared to previous plots, with SSI failing to capture high-stress events accurately. The threshold markers at 0.2 and 0.4 indicate the SSI is above the threshold and can detect high social stress, but compared to SSS is much lower, reinforcing concerns about SSI's sensitivity (Fig 3). The Bland-Altman plot illustrated a substantial negative bias of –0.468, which is nearly double the bias seen in Fig 2. The limits

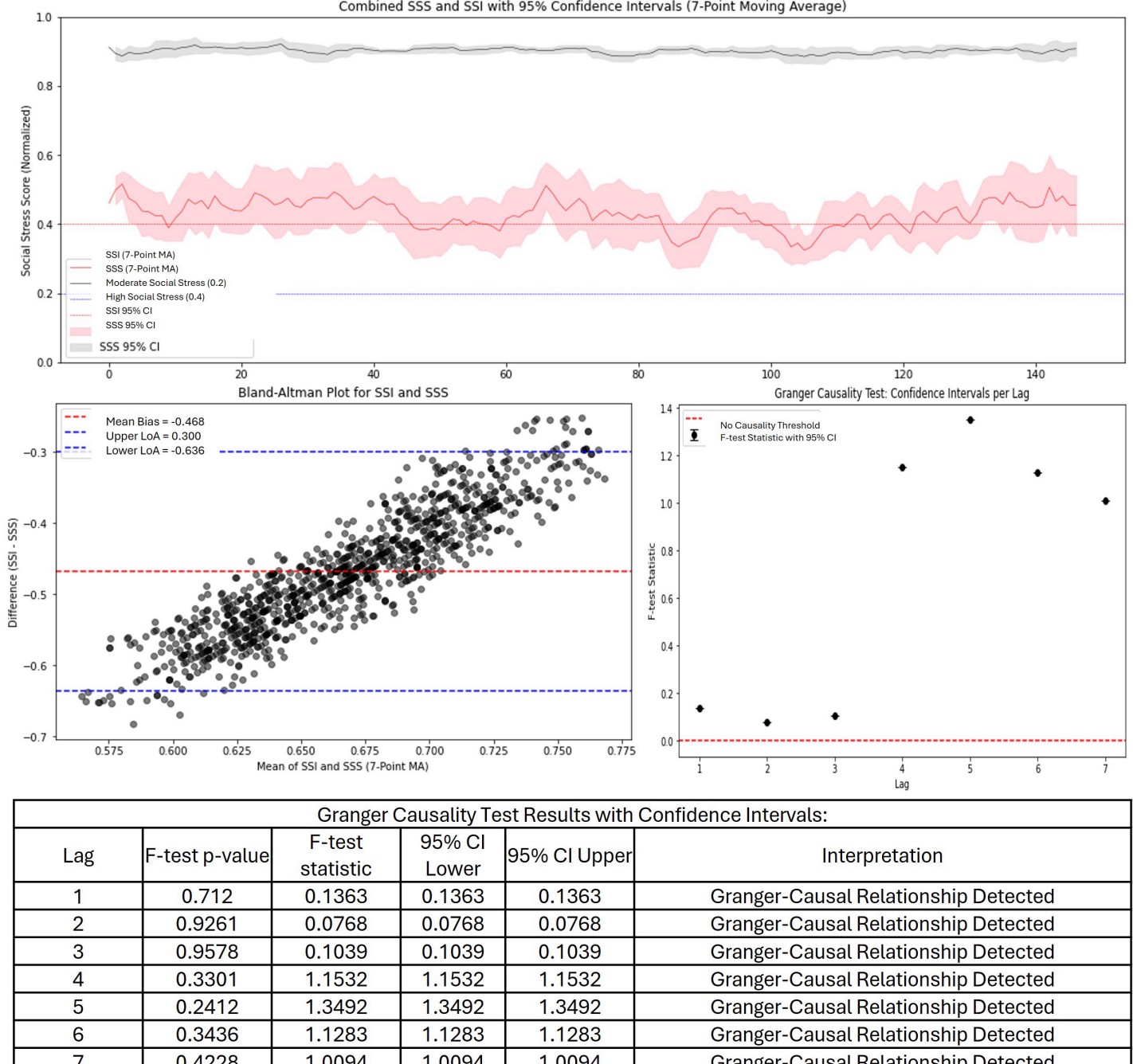

**Fig 3. SSI and SSS time series, Bland-Altman, and Granger causality for high stress microblogs.**

of agreement are also much wider, indicating high variability in how SSI compares to SSS. Despite these issues, the causality analysis actually shows more significant points above the causality threshold compared to previous analyses, suggesting that SSI still retains some predictive power for SSS, even though it underestimates it significantly in terms of intensity.

The SSI and SSS are designed to quantify and track social stress levels, but their comparative analysis reveals both strengths and weaknesses. Across the evaluations, SSI generally follows the trend of SSS but with varying degrees of intensity. The evaluation of a dataset with low social stress (Fig 1) show that SSI is a reasonable approximation of SSS, with a small bias and moderate agreement. This suggests that SSI successfully captures underlying stress dynamics, making it a potentially useful tool for tracking social stress in real-time of low stress content. Additionally, the causality tests indicate that SSI holds some predictive power, meaning it could be used for forecasting stress trends based on social media data. With moderate and high social stress data (Figs 2 and 3) SSI increasingly underestimates SSS, leading to a growing mismatch between the two measures. The negative bias worsens significantly, and the limits of agreement become wider, indicating more variability and inconsistency in how SSI aligns with expert-assessed SSS. The time-series analysis clearly shows that SSI fails to capture the full magnitude of social stress events, suggesting that it is either missing key stress indicators or over-smoothing data. While the causality analysis still supports some predictive capability. In all instances, social stress did lie above the threshold, even though there were discrepancies.

The SSI demonstrates several strengths that highlight its potential as a real-time monitoring tool for social stress dynamics. One of its key advantages is its ability to track general trends in social stress, as it aligns moderately well with the SSS in early evaluations. The predictive capability of SSI, as evidenced by causality tests, suggests that it can provide foresight into stress fluctuations, making it useful for forecasting emerging crises or infodemic trends. The inclusion of moving averages and confidence intervals stabilizes the indicator, reducing noise and improving its interpretability. These factors collectively suggest that SSI has the potential to be an effective real-time tool for tracking stress across social media platforms, particularly when calibrated correctly.

The SSI offers an empirically grounded proxy for detecting moments of elevated public distress, anxiety, and information pressure. By integrating sentiment analysis, topic models, and information-seeking behavior, the SSI has the potential to capture dynamic shifts in online conversation that reflect broader societal stressors. Additionally, the indicator can be trained on data to build a model that can be embedded on social media. While further calibration is necessary to account for linguistic nuance, demographic representation, and the evolving nature of digital discourse, the SSI remains a crucial early-warning tool for flagging high-stress narratives, misinformation spikes, and public reactions to crises. The SSI utility lies not in absolute precision but in its capacity to surface concerning trends that warrant closer expert analysis, policy attention, or health communication interventions - especially in the advent of infodemics, and other social listening related health tasks.

However, SSI also presents several critical weaknesses that limit its effectiveness in its current form. The most significant issue is its systematic underestimation of social stress, as evidenced by an increasingly negative bias in later evaluations. The Bland-Altman analysis indicates that as social stress levels increase, the gap between SSI and SSS widens, suggesting that SSI fails to capture high-stress events with sufficient sensitivity. This underestimation may be due to an over-reliance on sentiment analysis, which does not fully reflect crisis-driven stress, anxiety, misinformation surges, or heightened engagement around distressing topics.

In its current form, the SSI provides a composite score that signals elevated levels of stress-related discourse, however, its sensitivity could be significantly enhanced through the inclusion of topic contextualisation and differential weighting of constituent variables. By aligning the SSI more closely with the SSS—a rubric-based, expert-informed benchmark—the

indicator could more accurately reflect the intensity and nature of online social stress. Contextual calibration, such as assigning greater weight to misinformation or anxiety-laden content within high-impact topics, would allow the SSI to express stress more precisely, thereby improving its reliability for real-time monitoring and discourse flagging. Additionally, the widening limits of agreement indicate greater variability in how SSI aligns with SSS, raising concerns about its consistency and robustness as a social stress metric. The fixed threshold values for moderate and high stress may further limit its adaptability across different social contexts, reducing its effectiveness in detecting sudden surges in stress. Addressing these weaknesses requires recalibrating the weight of different SSI components, incorporating more dynamic modeling approaches, and validating the indicator against external datasets to enhance its reliability and sensitivity. Another limitation is the ISB values as certain topics might be more frequent during certain times of day, depending where in the world this is tracked.

## Conclusion

This study introduced the SSI as a computational approach to quantifying real-time social stress on digital platforms, integrating sentiment analysis (SA), subjectivity scores (SUB), and information seeking behavior (ISB). The validation of SSI against the expert-labeled SSS revealed both promising capabilities and key limitations. While SSI successfully tracked fluctuations in social stress and demonstrated moderate agreement with SSS in lower stress contexts, its performance deteriorated as social stress levels increased. The results showed increasing underestimation of stress in higher-intensity scenarios, with the bias widening significantly in later evaluations. Despite this limitation, causality tests suggested that SSI retains predictive power, indicating potential utility in early warning systems for detecting emerging stress related trends in social media discourse. The findings highlight that while SSI is a scalable and dynamic tool, further refinements are needed to enhance its sensitivity to extreme stress events and ensure greater alignment with expert assessments.

Future improvements should focus on recalibrating SSI's weighting of stress-related components, particularly enhancing its ability to capture misinformation surges, crisis-driven anxiety, and social engagement patterns. Additionally, adopting adaptive thresholds instead of fixed stress levels could increase SSI's flexibility across different social contexts and platforms. The incorporation of additional contextual signals—such as network effects, linguistic nuances (like cultural and linguistic adaptation), and external event tracking—may further improve accuracy and robustness. Furthermore, validation using real-world datasets and crisis scenarios will be critical in ensuring SSI's applicability for mental health monitoring, policy-making, and digital governance. Ultimately, this study contributes a novel computational framework for social stress detection, laying the foundation for future research aimed at developing more reliable and responsive infodemic intelligence systems.

## Author contributions

**Conceptualization:** Herkulaas Combrink.

**Data curation:** Herkulaas Combrink.

**Formal analysis:** Herkulaas Combrink.

**Investigation:** Herkulaas Combrink.

**Methodology:** Herkulaas Combrink.

**Project administration:** Herkulaas Combrink.

**Resources:** Herkulaas Combrink.

**Software:** Herkulaas Combrink.

**Supervision:** Herkulaas Combrink.

**Validation:** Herkulaas Combrink.

**Visualization:** Herkulaas Combrink.

**Writing – original draft:** Herkulaas Combrink.

**Writing – review & editing:** Herkulaas Combrink.

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
