## [Decision Letter · Decision Letter 0]

17 Jun 2025

PONE-D-25-06360The Social Stress Indicator (SSI)PLOS ONE

Dear Dr. Herkulaas MvE Combrink

Thank you for submitting your manuscript to PLOS ONE. After careful consideration, we feel that it has merit but does not fully meet PLOS ONE’s publication criteria as it currently stands. Therefore, we invite you to submit a revised version of the manuscript that addresses the points raised during the review process.

We look forward to receiving your revised manuscript.

Kind regards,

Maria José Nogueira, Ph.D.

Academic Editor

PLOS ONE

Journal Requirements:

Additional Editor Comments (if provided):

The study is timely and relevant, particularly in light of growing concerns about the impact of social media on mental health. It has the potential to offer an innovative perspective on social stress in the digital age.

However, certain aspects of the study could benefit from further development and refinement.

Reviewers' comments:

Reviewer's Responses to Questions

**Comments to the Author**

1. Is the manuscript technically sound, and do the data support the conclusions?

Reviewer #1: Yes

2. Has the statistical analysis been performed appropriately and rigorously? 

Reviewer #1: Yes

3. Have the authors made all data underlying the findings in their manuscript fully available?

Reviewer #1: Yes

4. Is the manuscript presented in an intelligible fashion and written in standard English?

Reviewer #1: Yes

5. Review Comments to the Author

Reviewer #1: I - Summary of the research and your overall impression

I congratulate the authors on the relevance of the study and the timeliness of the topic, with regard to the impact of social media on mental health, resulting in an innovative contribution to the analysis of social stress in the digital age. From this perspective, they present, in a well-structured approach, the proposal of a Social Stress Indicator (SSI) as a computational tool to quantify social stress in real time, integrating sentiment analysis (SA), subjectivity (SUB), and information seeking behaviour (ISB).

The study obtained ethical approval for the use of synthetic data, with no access to real individuals' data. The validation of the SSI was carried out through comparison with a Social Stress Score (SSS), grounded in psychological and behavioural indicators, namely, negative sentiment, anxiety expression, level of engagement, misinformation content and help seek behavior, validated by experts.

For a better understanding of the alignment between SSI and SSS, quantitative evaluation metrics were used, such as the Bland-Altman plot for agreement analysis and the Granger causality test for predictive capacity. The results prove the potential of SSI as a real-time social stress monitoring tool, but also its limitations in sensitivity and accuracy at high levels of social stress, underestimating high-intensity events.

It concludes by suggesting improvements in the refinement of this tool and values the contributions of this research to infodemic intelligence with relevance to the monitoring of mental health, health policies, and governance management.

II. Discussion of specific areas for improvement

A detailed analysis of this research allows me to make specific observations, as well as some suggestions and questions aimed at clarifying certain aspects of the methodological process. The analysis follows the structure of the article and concludes with a review of some “lapses” in the writing.

1. Introduction

• Considering the relevance of the post-pandemic period in studying the impact of social media on mental health, would it not be appropriate to prioritise research published after this period? It is noted that more than 50% of the references are from before 2020.

• It would be pertinent for the study to clarify the rationale for using synthetic data in your research.

• Regarding the “state of the art”, which gaps does your research seek to address?

2. Materials and Methods

• Lines 100-101: At present, there is a growing expectation that scientific knowledge should reach the general public and be accessible, thereby contributing to literacy and fostering an informed population. It would therefore be appropriate to explain the use of the three dimensions underpinning the Social Stress Indicator (SSI), namely sentiment analysis, subjectivity, and information-seeking behaviour. How were these dimensions/criteria identified or selected?

3. Ethics statement

• Lines 113-114: Data were generated using a Large Language Model (LLM). What methodological precautions did you use to make these data reliable?

4. SSS

• Lines 135-136: "Additionally, five expert raters applied the rubric to synthetic microblogs generated to reflect diverse stress-inducing contexts". Could you present the eligibility criteria for the expert evaluators? It would also be appropriate to provide further details on the evaluation procedures followed by these experts.

5. Results

• Lines 258-259: "SSI fails to capture high-stress events with sufficient sensitivity". Have you considered reflecting on the reasons behind the underestimation at high levels of stress, complementing your statistical analysis with qualitative approaches, such as the impact of cultural specificities?

• It would be appropriate to include a “Discussion” section in which the practical impact of the Social Stress Indicator is addressed — for example, its potential contributions to public policy, particularly in the field of mental health, in support of the 2030 Sustainable Development Goals, as well as the challenges associated with its implementation.

6. Conclusion

• Specify the main lines of future research. For example, have you considered the possibility of cultural and linguistic adaptation of this indicator?

7. Minor Comments (Article structure)

I identify some lapses in writing, namely:

• Line 103: Eliminate Social Stress Score, since in line 99, he has already presented, in full, the meaning of the acronym. On line 103, it must keep only "SSS".

• Line 109: An isolated "A" appears with no text. Is it a sentence that was cut and, by mistake, left the letter "A"? If yes, remove "A".

• Lines 128 and 141: Suggestion: Avoid that the chapter has the designation of Acronyms. For example, replace SSI and SSS with "Social Stress Indicator" and "Social Stress Score", respectively.

• Line 195: replace "for" with "For", as it is the beginning of a sentence. It must have been a mistake.

• Figures 1, 2 and 3 are barely noticeable and cannot be downloaded. Accessibility to them is suggested for a better analysis.

6. PLOS authors have the option to publish the peer review history of their article (what does this mean?). If published, this will include your full peer review and any attached files.

Reviewer #1: No

---

## [Author Response · Author response to Decision Letter 1]

6 Jul 2025

Reviewer Comment Feedback

Comment: Introduction: Considering the relevance of the post-pandemic period in studying the impact of social media on mental health, would it not be appropriate to prioritise research published after this period? It is noted that more than 50% of the references are from before 2020.

A:That number reduced with the introduction of new references above 2024, and it must be noted that the earlier references are context specific to ideas in the field of which academic reference was given to the source of the ideas in those instances.

Comment: Introduction: It would be pertinent for the study to clarify the rationale for using synthetic data in your research.

A: This context was added in the methodology as synthetic data generated was specific to generic, rather than nuance content for the experimentation. As this study was conducted specific to SSI and its context to the indicator, future studies relate to the context specific research trained and used on real-world microblogs from a specific online discourse.

Comment: Introduction: Regarding the “state of the art”, which gaps does your research seek to address?

A:Added contextualisation and the relevant sources into what online social stress lacks, and what an SSI does differently to traditional approaches.

Comment: Materials and Methods: Lines 100-101: At present, there is a growing expectation that scientific knowledge should reach the general public and be accessible, thereby contributing to literacy and fostering an informed population. It would therefore be appropriate to explain the use of the three dimensions underpinning the Social Stress Indicator (SSI), namely sentiment analysis, subjectivity, and information-seeking behaviour. How were these dimensions/criteria identified or selected?

A:A detailed and referenced (using the latest relevant literature from 2024) was added to the section.

Comment: Ethics statement: Lines 113-114: Data were generated using a Large Language Model (LLM). What methodological precautions did you use to make these data reliable?

A:Context was added over the prompt contextualisation, as well as that every synthetic datapoint produced was vetted by the principal investigator.

Commnet: SSS: Lines 135-136: "Additionally, five expert raters applied the rubric to synthetic microblogs generated to reflect diverse stress-inducing contexts". Could you present the eligibility criteria for the expert evaluators? It would also be appropriate to provide further details on the evaluation procedures followed by these experts.

A: Context explained in revised version.

Comment: Results: Lines 258-259: "SSI fails to capture high-stress events with sufficient sensitivity". Have you considered reflecting on the reasons behind the underestimation at high levels of stress, complementing your statistical analysis with qualitative approaches, such as the impact of cultural specificities?

A: Absolutely, and part of the SSI is the contextual relevance. I added a section explaining that the indicator would expressed elevated stress levels as compared to the Social stress score if weighting as well as topic contextualisation was included in the indicator.

Comment: Results: It would be appropriate to include a “Discussion” section in which the practical impact of the Social Stress Indicator is addressed — for example, its potential contributions to public policy, particularly in the field of mental health, in support of the 2030 Sustainable Development Goals, as well as the challenges associated with its implementation.

A: Results session is now “Results and Discussion”. A section was added on the practical impact of the SSI, but not based on the example provided. Given that the SSI would be best suited for social media, the practical implementation of the SSI for social media was recommended.

Comment: Conclusion: Specify the main lines of future research. For example, have you considered the possibility of cultural and linguistic adaptation of this indicator?

A: Added in the recommendation “… linguistic nuances (like cultural and linguistic adaptation) …”

Comment: Minor Comments (Article structure): Line 103: Eliminate Social Stress Score, since in line 99, he has already presented, in full, the meaning of the acronym. On line 103, it must keep only "SSS".

A: Changed.

Comment: Minor Comments (Article structure): Line 109: An isolated "A" appears with no text. Is it a sentence that was cut and, by mistake, left the letter "A"? If yes, remove "A".

A: Changed.

Comment: Minor Comments (Article structure): Lines 128 and 141: Suggestion: Avoid that the chapter has the designation of Acronyms. For example, replace SSI and SSS with "Social Stress Indicator" and "Social Stress Score", respectively.

A: Changed.

Comment: Minor Comments (Article structure): Line 195: replace "for" with "For", as it is the beginning of a sentence. It must have been a mistake.

A: Changed.

Comment: Minor Comments (Article structure): Figures 1, 2 and 3 are barely noticeable and cannot be downloaded. Accessibility to them is suggested for a better analysis.

A:Updated figures for quality. The download would be ready upon publication and not during the draft.

---

## [Editor Report · Decision Letter 1]

8 Jul 2025

The Social Stress Indicator (SSI)

PONE-D-25-06360R1

Dear Dr. Herkulaas MvE Combrink,

We’re pleased to inform you that your manuscript has been judged scientifically suitable for publication and will be formally accepted for publication once it meets all outstanding technical requirements.

Kind regards,

Maria José Nogueira, Ph.D.

Academic Editor

PLOS ONE

Additional Editor Comments (optional):

The authors made the changes recommended by the reviewers, increasing its clarity and robustness. The manuscript can be accepted.
---

## [Editor Report · Acceptance letter]

PONE-D-25-06360R1

PLOS ONE

Dear Dr. Combrink,

I'm pleased to inform you that your manuscript has been deemed suitable for publication in PLOS ONE. Congratulations! Your manuscript is now being handed over to our production team.

Kind regards,

on behalf of

Professor Maria José Nogueira

Academic Editor

PLOS ONE